# Symmetry-protected exceptional and nodal points in non-Hermitian systems

Sharareh Sayyad[1]⋆, Marcus Stålhammar[2,3], Lukas Rødland[2] and Flore K. Kunst[1]

**1** Max Planck Institute for the Science of Light, Staudtstraße 2, 91058 Erlangen, Germany
**2** Department of Physics, Stockholm University,
AlbaNova University Center, 106 91 Stockholm, Sweden
**3** Nordita, KTH Royal Institute of Technology and Stockholm University,
Hannes Alfvéns väg 12, SE-106 91 Stockholm, Sweden

⋆ sharareh.sayyad@mpl.mpg.de

## Abstract

One of the unique features of non-Hermitian (NH) systems is the appearance of NH degeneracies known as exceptional points (EPs). The extensively studied defective EPs occur when the Hamiltonian becomes non-diagonalizable. Aside from this degeneracy, we show that NH systems may host two further types of non-defective degeneracies, namely, non-defective EPs and ordinary (Hermitian) nodal points. The non-defective EPs manifest themselves by i) the diagonalizability of the NH Hamiltonian at these points and ii) the non-diagonalizability of the Hamiltonian along certain intersections of these points, resulting in instabilities in the Jordan decomposition when approaching the points from certain directions. We demonstrate that certain discrete symmetries, namely parity-time, parity-particle-hole, and pseudo-Hermitian symmetry, guarantee the occurrence of both defective and non-defective EPs. We extend this list of symmetries by including the NH time-reversal symmetry in two-band systems. Two-band and four-band models exemplify our findings. Through an example, we further reveal that ordinary nodal points may coexist with defective EPs in NH models when the above symmetries are relaxed.



# 1  Introduction

Despite violating the axioms of quantum mechanics, non-Hermitian (NH) Hamiltonians offer compelling descriptions for numerous interacting/open systems in various fields of physics [1–7]. The underlying physics of these effective Hamiltonians goes beyond the realm of Hermitian physics and has been immensely studied lately [8–23]. Aside from unraveling rich physics, the properties of NH systems are well-reflected in abstract mathematical frameworks, including homotopy theory [24–28] and K-theory [29–31]. These frameworks provide a reliable toolbox to understand the exotic properties of NH systems and distinguish their behavior from Hermitian physics.

One notable distinction between NH Hamiltonians and their Hermitian counterparts is the appearance of exceptional points (EPs) as NH degeneracies. While one generally should satisfy $n-1$ complex constraints to realize EPs of order $n$ (EP$n$s), at which the Hamiltonian casts a Jordan block, recent studies have shown that the presence of certain discrete symmetries, such as parity-time ($\mathcal{PT}$), parity-particle-hole ($\mathcal{CP}$), or pseudo-Hermiticity (psH) symmetry, reduces the total number of constraints [32–43]. Although these studies mainly focused on characterizing symmetry-induced restrictions on defective EPs, drawing a link between discrete symmetries and non-defective degeneracies associated with the NH Hamiltonian beyond case studies has received little attention. Furthermore, while there is a near consensus on calling defective degeneracies [44, 45] defective EPs, non-defective degeneracies in NH systems are referred to as diabolic points [46–49], Fermi points [50], Dirac points and vertex points [51].[1] These non-defective degeneracies are different in nature. While both Dirac points and vertex points might appear in the vicinity of defective EPs, diabolic points and Fermi points are similar to (Hermitian) ordinary nodal points (ONPs).

Recalling their mathematical origin, EPs were introduced by Kato as isolated singularities of systems depending on one complex variable [52], and they were recently classified into type I EPs and type II EPs [2]. EPs of type I are degeneracies with or without algebraic singularities reminiscent of defective EPs and ONPs. In contrast, type II EPs are defined as points in the complex plane where the Jordan normal form is unstable, i.e., the eigenprojectors have a pole. While the appearance and existence of degeneracies reminiscent of type I EPs have been studied extensively, type II EPs have hitherto been overlooked in the literature.

In this work, we introduce a natural extension of type II EPs to higher dimensions, dubbed *non-defective EPs*. We present that the correct criterion for detecting non-defective EPs is the form of the Hamiltonian matrix in the *vicinity* of these degenerate points: Non-defective EPs are surrounded by defective EPs in certain intersections such that the Hamiltonian matrix casts a Jordan block along directions where defective EPs reside. Furthermore, we show that the Jordan decomposition is singular at these points when approaching the points from certain

---

[1]An exception is considered in Ref. [51] where 'hybrid points' have also been introduced based on the asymptotic dispersion relations close to defective degeneracies. Note that branch cuts do not terminate at hybrid points [50].

| | Defective degeneracies[●] | Non-defective degeneracies[●] | Ordinary nodal points[●] |
|---|---|---|---|
| Constraints to emerge | Third columns in Tables 1-3 | Fourth columns in Tables 1-3 | Null traceless part of the Hamiltonian |
| Intuitive identification | $\sqrt[m]{k}$ dispersion with $1 < m \leq n$ | Vicinity to defective EPs | No defective EPs in the vicinity |
| Exemplified two-band spectrum | Re[ε]    Im[ε]    k   k | Re[ε]    Im[ε]    k   k | Re[ε]    Im[ε]    k   k |

Figure 1: Schematic illustration of three different degeneracies in non-Hermitian systems. For each degeneracy, we list the required constraints for the occurrence of degeneracies, an intuitive way to identify them, and an example of the dispersion relation of such degeneracies in two-level systems. Here, $n$ denotes the order of the degeneracy.

directions, emphasizing that non-defective EPs indeed are reminiscent of the type II EPs in Ref. [2]. To characterize the role of symmetries in witnessing NH degeneracies, we study the coexistence of defective and non-defective EP$ns$ in two-, three- and four-band models in the presence of psH, $\mathcal{PT}$ or $\mathcal{CP}$ symmetry. Additionally, we show that symmetry-protected non-defective EP2s may also appear in models with non-Hermitian time-reversal symmetry. We include a two- and four-band example to highlight our findings. Finally, we find that defective EPs may coexist with ONPs instead of non-defective EPs when lifting symmetry constraints. We illustrate this finding with a fine-tuned example. Our findings are summarized in Fig. 1.

## 2 Symmetry-stabilized (non-)defective EPs

A generic $n$-band Hamiltonian can be decomposed as

$$\mathcal{H} = d_\mu \Upsilon^\mu, \tag{1}$$

where $\mu \in \{0, \cdots, n^2 - 1\}$, $d_\mu$ are continuously differentiable complex-valued functions of the lattice momentum $\mathbf{k}$, $\Upsilon^0$ denotes the identity matrix of order $n$ and $\Upsilon$ is the basis of the $SU(n)$ group, which consists of three Pauli matrices when $n = 2$, eight Gell-Mann matrices when $n = 3$, and fifteen generalized Gell-Mann matrices when $n = 4$, see Appendix A. The Hamiltonian $\mathcal{H}$ displays $\mathcal{PT}$ symmetry with generator $\mathcal{PT}$, $\mathcal{CP}$ symmetry with generator $\mathcal{CP}$, or psH with generator $\varsigma$, if it satisfies one of the following relations, namely,

$$\mathcal{PT}: \quad \mathcal{H}(\mathbf{k}) = (\mathcal{PT}) \mathcal{H}^*(\mathbf{k})(\mathcal{PT})^{-1}, \tag{2}$$

$$\mathcal{CP}: \quad \mathcal{H}(\mathbf{k}) = -(\mathcal{CP}) \mathcal{H}^*(\mathbf{k})(\mathcal{CP})^{-1}, \tag{3}$$

$$\text{psH}: \quad \mathcal{H}(\mathbf{k}) = \varsigma \mathcal{H}^\dagger(\mathbf{k})\varsigma^{-1}. \tag{4}$$

These symmetry considerations reduce the number of non-zero $d_\mu$ values. To be precise, for each basis matrix $\Upsilon^\mu$, only either the real or imaginary part of $d_\mu$ remains non-zero. This means only one real-valued function $d_\mu$ survives for each dimension of $\Upsilon^\mu$ [42]. Trivial band touching points occur when the traceless part of $\mathcal{H}$ becomes a Null matrix ($[0]_{n \times n}$), i.e., all of the non-zero $d_\mu$ values for $\mu > 0$ must vanish, which means that one needs to satisfy $n^2 - 1$ real constraints. For $n = 2, 3, 4$, we have collected these $d_\mu$'s for each symmetry operation alongside a choice for its generator in Tables 1, 2, and 3, respectively.

It was shown in Ref. [42] that the $n-1$ complex constraints to find defective EP$ns$ can be expressed in terms of the traces and the determinant of $\mathcal{H}$, which for two-, three-, and four-band models, respectively, read

$$n=2: \quad \eta^{2b} = \text{tr}[\mathcal{H}]^2 - 4\det[\mathcal{H}], \tag{5}$$

$$n=3: \begin{cases} \eta^{3b} = \frac{1}{2}\left(\text{tr}[\mathcal{H}]^2 - 3\,\text{tr}[\mathcal{H}^2]\right), \\ \nu^{3b} = \frac{1}{2}\left(54\det[\mathcal{H}] - 5\,\text{tr}[\mathcal{H}]^3 + 9\,\text{tr}[\mathcal{H}]\,\text{tr}[\mathcal{H}^2]\right), \end{cases} \tag{6}$$

$$n=4: \begin{cases} \eta^{4b} = -3ac + b^2 + 12d, \\ \nu^{4b} = 27a^2d - 9abc + 2b^3 - 72bd + 27c^2, \\ \kappa^{4b} = a^3 - 4ab + 8c, \end{cases} \tag{7}$$

where

$$a = \text{tr}[\mathcal{H}], \tag{8}$$

$$b = \frac{(\text{tr}[\mathcal{H}])^2 - \text{tr}[\mathcal{H}^2]}{2}, \tag{9}$$

$$c = \frac{\text{tr}[\mathcal{H}]^3 - 3\,\text{tr}[\mathcal{H}]\,\text{tr}[\mathcal{H}^2] + 2\,\text{tr}[\mathcal{H}^3]}{6}, \tag{10}$$

$$d = \det[\mathcal{H}]. \tag{11}$$

We refer to Appendix B for details on how to derive these expressions.

In the presence of $\mathcal{PT}$, $\mathcal{CP}$, or psH symmetry, some of these constraints are automatically satisfied, leaving us with exactly $n-1$ real constraints, cf. Tables 1, 2 and 3 [42]. It is notable that at trivial solutions ($\mathbf{d}=0$), all traces and the determinant of $\mathcal{H}$ acquire zero values, and subsequently, the constraints in Eqs. (5)-(7) are also satisfied. The trivial solutions mark non-defective EP$ns$ with the binding signature that $\mathcal{H}$ is diagonalizable at these points.[2] Perturbing the system away from these non-defective points along intersections at which the constraints vanish brings the Hamiltonian into a non-diagonalizable structure. We set this behavior as a criterion to detect non-defective EPs. We note that this is opposed to the situation in which trivial solutions are isolated, and thus, band touching points behave similarly to Hermitian degeneracies, i.e., ONPs.

In summary, we thus notice that NH systems may exhibit three different eigenvalue degeneracies, schematically depicted in Fig. 1. Formally, the different degeneracies are defined as follows:

- **Defective EPs:** Eigenvalue degeneracies appearing for non-zero $\mathbf{d}$ at which also eigenvectors coalesce, making the Hamiltonian cast a Jordan block form.

- **ONPs:** Eigenvalue degeneracies appearing for trivial $\mathbf{d}=0$, which are isolated from defective EPs. The Hamiltonian is diagonalizable at ONPs and remains diagonalizable in any infinitesimally small but finite neighborhood around the region $\mathbf{d}=0$.

- **Non-Defective EPs:** Eigenvalue degeneracies appearing for trivial $\mathbf{d}=0$ located in the direct vicinity of defective EPs, usually comprising intersections of defective EPs. The Hamiltonian is diagonalizable at non-defective EPs but casts a Jordan block in certain directions away from, yet arbitrarily close to, the non-defective EPs. Consequently, when approaching the non-defective EP along any non-diagonalizable direction, the Jordan decomposition becomes singular.

---

[2]We note that in the absence of $\mathcal{PT}$, $\mathcal{CP}$, or psH symmetry, trivial solutions describe one of the non-defective degeneracies, i.e., non-defective EP$ns$, or ordinary nodal points.

Table 1: Summarized symmetries, their generators, and their associated constraints to find defective and non-defective EP2s in two-band systems.

| Symm. | Generator | Constr. def. EP2s | Constr. non-def. EP2s |
|---|---|---|---|
| $\mathcal{PT}$ | $\mathbb{1}$ | $\eta_R^{2b} = 0$ | $d_{xR} = d_{yI} = d_{zR} = 0$ |
| $\mathcal{CP}$ | $\mathbb{1}$ | $\eta_R^{2b} = 0$ | $d_{xI} = d_{yR} = d_{zI} = 0$ |
| psH | $\mathrm{adiag}[1,1]$ | $\eta_R^{2b} = 0$ | $d_{xR} = d_{yI} = d_{zI} = 0$ |
| TRS$^\dagger$ | $\mathrm{adiag}[1,-1]$ | $\eta_R^{2b} = \eta_I^{2b} = 0$ | $d_{xa} = d_{ya} = d_{za} = 0$ |

Here we use either $d_j = d_{jR} + \mathrm{i}d_{jI}$ or $d_j = d_{js} + d_{ja}$, where $d_{js}(d_{ja})$ is (anti-)symmetric with respect to $\mathbf{k} \to -\mathbf{k}$. $\eta^{2b}$ is given in Eq. (5) with $\eta^{2b} = \eta_R^{2b} + \mathrm{i}\eta_I^{2b}$.

Table 2: Summarized symmetries, their generators, and their associated constraints to find defective and non-defective EP3s in three-band systems.

| Symm. | Generator | Constr. def. EP3s | Constr. non-def. EP3s |
|---|---|---|---|
| $\mathcal{PT}$ | $\mathrm{diag}[1,-1,1]$ | $\eta_R^{3b} = v_R^{3b} = 0$ | $d_{1R} = d_{2I} = d_{3R} = d_{4I} = 0$ $d_{5R} = d_{6I} = d_{7R} = d_{8R} = 0$ |
| $\mathcal{CP}$ | $\mathrm{diag}[1,-1,1]$ | $\eta_R^{3b} = v_I^{3b} = 0$ | $d_{1I} = d_{2R} = d_{3I} = d_{4R} = 0$ $d_{5I} = d_{6R} = d_{7I} = d_{8I} = 0$ |
| psH | $\mathrm{diag}[1,-1,1]$ | $\eta_R^{3b} = v_R^{3b} = 0$ | $d_{1I} = d_{2R} = d_{3I} = d_{4I} = 0$ $d_{5R} = d_{6I} = d_{7R} = d_{8R} = 0$ |

Here we use $d_j = d_{jR} + \mathrm{i}d_{jI}$. Complex valued $\eta^{3b}$ and $v^{3b}$ constraints are given in Eq. (6) with $\alpha^{3b} = \alpha_R^{3b} + \mathrm{i}\alpha_I^{3b}$ for $\alpha \in \{\eta, v\}$.

The diagonalizability of the Hamiltonian at non-defective EPs enables us to map our NH Hamiltonians into their Hermitian counterparts with nodal points. In addition, having $n^2 - 1$ non-zero $d_\mu$'s as in Hermitian systems enforces non-defective EPs to always appear in pairs in the Brillouin zone. This statement originates from the Poincaré-Hopf theorem [53], as the number of non-zero $d_\mu$ functions equals the dimension of the vector space ($n^2 - 1$), cf. last columns in Tables 1, 2, and 3.[3] Consequently, these non-defective EPs are topological in the same sense as, e.g., Weyl points in Hermitian systems, and can be classified by Chern numbers.

Aside from $\mathcal{PT}$, $\mathcal{CP}$, and psH symmetries, a particular non-Hermitian time-reversal symmetry, known as TRS$^\dagger$, in two-band systems may also give rise to realizing non-defective EPs. To evidently see this behavior, we recall that respecting TRS$^\dagger$ symmetry imposes $\mathcal{H}(-\mathbf{k}) = \mathcal{C}_+ \mathcal{H}^\mathrm{T}(\mathbf{k})\mathcal{C}_+^\dagger$ [31]. This non-(momentum)-local transformation does not reduce the number of non-vanishing (real/imaginary) parts of $d_\mu$. However, when $C_+ C_+^* = -\mathbb{1}$, e.g., $\mathcal{C}_+ = \mathrm{i}\sigma_y$, it enforces all symmetric parts of $d_\mu$ to become zero. We further know that at the time-reversal invariant momenta (TRIM), functions that are anti-symmetric with respect to $\mathbf{k}$ vanish. Therefore, at $\mathbf{k}_\mathrm{TRIM}$, both real and imaginary parts of anti-symmetric $d_\mu$ functions become zero, which gives rise to the observation of non-defective EPs in the spectra of the two-band TRS$^\dagger$-symmetric Hamiltonian $\mathcal{H}$ in 3D, cf. Table 1. The non-defective EPs protected by TRS$^\dagger$ are also topological in the sense that there exist non-trivial loops around them. Yet, they are different from those arising in, e.g., $\mathcal{PT}$-symmetric systems, as the TRIM are fixed points. Hence, non-defective EPs are stationary in momentum space, and a non-defective EP cannot be annihilated by merging together with a non-defective EP of opposite topological charge. This means that there does not exist a unified notion of topological invariants for non-defective EPs, but the classification depends on the present symmetry.

---

[3]This can also be deduced from the Nielsen-Ninomiya Theorem. See also Ref. [50].

Table 3: Summarized symmetries, their generators, and their associated constraints to find defective and non-defective EP4s in four-band systems.

| Symm. | Generator | Constr. def. EP4s | Constr. non-def. EP4s |
|-------|-----------|-------------------|------------------------|
| $\mathcal{PT}$ | $\text{diag}[1,-1,1,-1]$ | $\eta_R^{4b} = \nu_R^{4b} = \kappa_R^{4b} = 0$ | $d_{1R} = d_{2I} = d_{3R} = d_{4R} = 0$ <br> $d_{5I} = d_{6R} = d_{7I} = d_{8R} = 0$ <br> $d_{9I} = d_{10I} = d_{11R} = d_{12I} = 0$ <br> $d_{13R} = d_{14R} = d_{15R} = 0$ |
| $\mathcal{CP}$ | $\text{diag}[1,-1,1,-1]$ | $\eta_R^{4b} = \nu_R^{4b} = \kappa_I^{4b} = 0$ | $d_{1I} = d_{2R} = d_{3I} = d_{4I} = 0$ <br> $d_{5R} = d_{6I} = d_{7R} = d_{8I} = 0$ <br> $d_{9R} = d_{10R} = d_{11I} = d_{12R} = 0$ <br> $d_{13I} = d_{14I} = d_{15I} = 0$ |
| psH | $\text{diag}[1,-1,1,-1]$ | $\eta_R^{4b} = \nu_R^{4b} = \kappa_R^{4b} = 0$ | $d_{1I} = d_{2R} = d_{3I} = d_{4I} = 0$ <br> $d_{5R} = d_{6I} = d_{7I} = d_{8R} = 0$ <br> $d_{9I} = d_{10I} = d_{11R} = d_{12I} = 0$ <br> $d_{13R} = d_{14R} = d_{15R} = 0$ |

Here we use $d_j = d_{jR} + \mathrm{i}d_{jI}$. Complex valued $\eta^{4b}$, $\nu^{4b}$ and $\kappa^{4b}$ constraints are given in Eq. (7) with $\alpha^{4b} = \alpha_R^{4b} + \mathrm{i}\alpha_I^{4b}$ for $\alpha \in \{\eta, \nu, \kappa\}$.

Before moving on to examples, we note that the different number of constraints that need to be satisfied to find symmetry-protected defective and non-defective EP*ns* also result in a different codimension of these EPs. Here, the codimension is given by the difference between the total dimension of the system and the dimension of the exceptional feature. Equivalently, the codimension corresponds precisely to the number of non-vanishing constraints. In particular, while the presence of $\mathcal{PT}$, $\mathcal{CP}$ and psH symmetries reduces the number of real constraints for finding defective EP*ns* to $n-1$, the number of real constraints to detect non-defective EP*ns* is $n^2 - 1$. As a consequence, in the case of $n = 2$, defective EP2s have codimension *one*, whereas non-defective EP2s have codimension *three*. Therefore, the latter appear as points in three-dimensional systems, whereas defective EP2s appear as two-dimensional surfaces. For the TRS$^\dagger$ invariant two-band model, the codimension of defective EP2s is *two*, and hence the defective EP2s are curves connected at the TRIMs.

**Examples for the coexistence of defective and non-defective EPs.**

We start with introducing a two-band $\mathcal{PT}$-symmetric Weyl-like tight-binding model described by

$$\begin{aligned}
\mathcal{H}_{\mathcal{PT}}^{2b} &= d_0 \Upsilon^0 + d_{xR} \Upsilon^1 + \mathrm{i}d_{yI} \Upsilon^2 + d_{zR} \Upsilon^3 \\
&= 2\lambda_0 \sin(k_x) \Upsilon^0 + 2t \sin(k_x) \Upsilon^1 + 2t \sin(k_y) \Upsilon^3 \\
&\quad + \mathrm{i}\{2t \cos(k_z) + 2V[2 - \cos(k_x) - \cos(k_y)]\} \Upsilon^2.
\end{aligned} \tag{12}$$

Here $\lambda_0$, $t$ and $V$ are real-valued parameters. The real and imaginary parts of the band structure are shown in Figs. 2(a) and (b), respectively. Non-defective EPs appear when all components of the Hamiltonian, except $d_0$, vanish. More specifically, these degeneracies emerge when the solutions of $d_{xR} = 0$ (at $k_x = n\pi$ with $n \in \mathbb{Z}$), $d_{yI} = 0$ (orange curves in Fig. 2(c)), and $d_{zR} = 0$ (grey line) intersect. Red points at $\mathbf{k} = (0, 0, \pm\pi/2)$ in Fig. 2(c) exemplify such solutions. Note that the criterion for detecting non-defective EPs is satisfied for the red points in Fig. 2(c) as they are surrounded by defective EPs (dashed blue curves), residing on $\eta_R^{2b} = 0$, where $\eta_R^{2b}$ is the real part of $\eta^{2b}$ in Eq. (5).

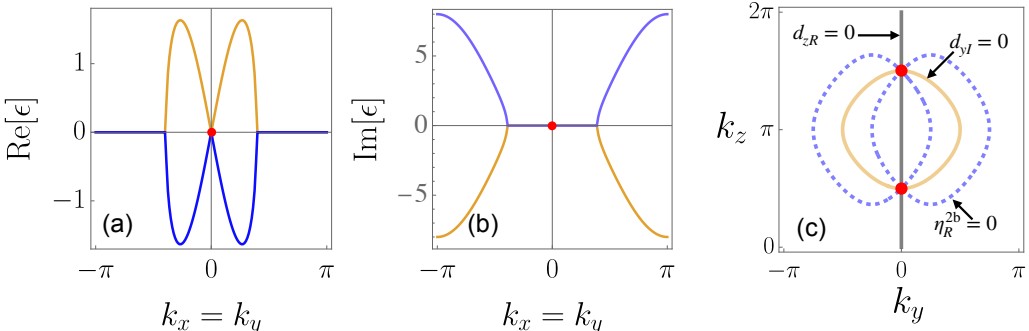

Figure 2: Real (a) and imaginary (b) parts of spectra for Eq. (12) along $k_x = k_y$ with $k_z = \pi/2$. Red points indicate non-defective EPs. Panel (c) displays solutions of $d_{yI} = 0$ (orange), $d_{zR} = 0$ (gray), and $\eta_R^{2b} = 0$ (dashed blue) at $k_x = 0$, which is a solution to $d_{xR} = 0$. Red points at $\mathbf{k} = (0, 0, \pm\pi/2)$ indicate the intersection between solutions to $\eta_R^{2b} = d_{yI} = d_{zR} = 0$, and are non-defective EPs. Here we set $t = V = 1$, and $\lambda_0 = 0$.

Let us now investigate how the dispersion and eigenvectors of $\mathcal{H}_{\mathcal{PT}}^{2b}$ behave at the non-defective EP. Around $\mathbf{k} = (0, 0, \pi/2) := \mathbf{k}_{\mathrm{NDEP}}$, the corresponding eigensystem reads,

$$\epsilon_{\mathcal{PT}}^{2b,\pm} = 2\left[ \lambda_0 \tilde{k}_x \pm \sqrt{t^2\left(\tilde{k}_x^2 + \tilde{k}_y^2 - \tilde{k}_z^2\right)}\, \right], \tag{13}$$

$$\psi_{\mathcal{PT}}^{2b,\pm} = \begin{pmatrix} \dfrac{t\tilde{k}_y \pm \sqrt{t^2\left(\tilde{k}_x^2 + \tilde{k}_y^2 - \tilde{k}_z^2\right)}}{\tilde{k}_x + \tilde{k}_z} \\ 1 \end{pmatrix}, \tag{14}$$

where we set $\tilde{\mathbf{k}} = \mathbf{k} - \mathbf{k}_{\mathrm{NDEP}}$. First, we note that the dispersion is linear in momentum, evident in Fig. 3. We further note the special behavior in the directions of the defective EPs, indicated by flat lines along $\tilde{k}_x = \pm\tilde{k}_z$ in Fig. 3(c). This differs significantly from the dispersion behavior around defective EPs, which is always fractional in momentum in two-band models. Second, we interestingly observe that the coalescence of the eigenvectors depends on which direction we approach the non-defective EP. Approaching the non-defective EP along the defective EPs,

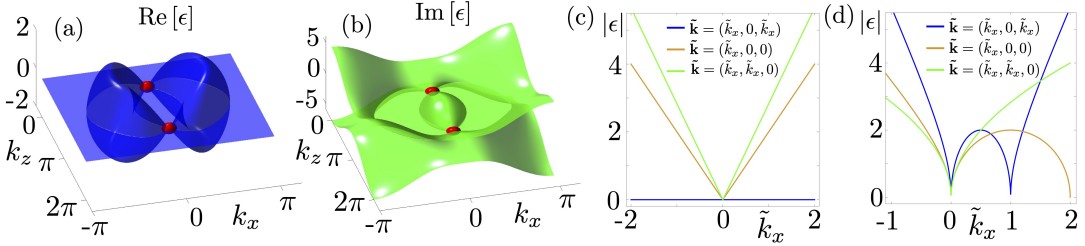

Figure 3: Real (a) and imaginary (b) part of band structure for $\mathcal{H}_{\mathcal{PT}}^{2b}$ in Eq. (12) at $k_y = 0$ along $k_x$ and $k_z$. Red points indicate non-defective EPs at $\mathbf{k} = (0, 0, \pm k_0)$. Panel (c) displays the linear dependence on the momentum of the absolute value of the eigenvalues at the non-defective EP $\mathbf{k} = (0, 0, \pi/2)$ in different cuts of momentum space. We notice flat directions along the defective EP. To contrast this, panel (d) shows the fractional behavior of the absolute value of the eigenvalues around the defective EP at $\mathbf{k} = (\pi/2, 0, \pi/2)$ in the same cuts. Here we set $t = V = 1$, and $\lambda_0 = 0$.

defined by $d_{xR}^2 + d_{zR}^2 = d_{yI}^2$, yields,

$$\lim_{\tilde{k}_x \to 0} \lim_{\tilde{k}_y \to \sqrt{\tilde{k}_z^2 - \tilde{k}_x^2}} \psi_{\mathcal{PT}}^{2b,\pm} = \begin{pmatrix} \sqrt{\tilde{k}_z^2} \\ \tilde{k}_z \\ 1 \end{pmatrix}. \tag{15}$$

When finally going to the non-defective EP, corresponding to further take the limit $\tilde{k}_z \to 0$, we can conclude that the eigenvectors seem to coalesce at the non-defective EP. Thus, when approached along the defective EPs, the origin of momentum space seems to comprise a defective point. If we instead approach the non-defective EP as, e.g.,

$$\lim_{\tilde{k}_y \to 0} \lim_{\tilde{k}_z \to 0} \psi_{\mathcal{PT}}^{2b,\pm} = \begin{pmatrix} \pm \frac{\sqrt{-t^2 \tilde{k}_x^2}}{t \tilde{k}_x} \\ 1 \end{pmatrix}, \tag{16}$$

we obtain two linearly independent eigenvectors even when taking the final limit $\tilde{k}_x \to 0$. This behavior is also reflected in the Jordan decomposition of $\mathcal{H}_{\mathcal{PT}}^{2b}$ on the exceptional surface around $\mathbf{k}_{\text{NDEP}}$ given by $\mathcal{H}_{\mathcal{PT}}^{2b} = SJS^{-1}$ with

$$S = \begin{pmatrix} \frac{\sqrt{\tilde{k}_z^2 - \tilde{k}_x^2}}{\tilde{k}_x + \tilde{k}_z} & \frac{1}{2t(\tilde{k}_x + \tilde{k}_z)} \\ 1 & 0 \end{pmatrix}, \tag{17}$$

$$J = \begin{pmatrix} 2\lambda_0 \tilde{k}_x & 1 \\ 0 & 2\lambda_0 \tilde{k}_x \end{pmatrix}. \tag{18}$$

The second column in the transformation matrix ($S$) exhibits a singularity when $\tilde{k}_x = -\tilde{k}_z$, along the defective EPs. Since there is a rotational symmetry on the exceptional surface, we can perform a coordinate change to move away from this singularity as long as $\tilde{k}_x$ and $\tilde{k}_z$ are non-zero. Thus, apart from $\tilde{k}_x = \tilde{k}_z = 0$, this amounts to a coordinate singularity. Exactly at the non-defective EP, however, there is a true singularity. Hence, the Jordan decomposition is unstable at the non-defective EPs.

As a comparison, we note that the band disperses like the square root of the momentum components around defective EPs, displayed in Fig. 3(d). Furthermore, the eigenvector coalescence at these points is not dependent on the direction from which the point is approached. In this sense, eigenvector coalescence is not a local system property in momentum space. Similar conclusions were recently made in Ref. [54].

We further note that the defective EPs separate two regions in the real part of the spectrum, where $\text{Re}[\Delta \epsilon] = 0$ and $\text{Re}[\Delta \epsilon] \neq 0$ with $\Delta \epsilon$ being the difference between the two energy bands as shown in Fig. 2(a). Regions where $\text{Re}[\Delta \epsilon] = 0$ are sometimes referred to as NH bulk real-Fermi states, which merely appear in NH systems [8]. In Appendix C, we show that besides these bulk Fermi states, this model also hosts states on the boundary. Therefore, there is a coexistence between defective and non-defective EPs as well as between bulk Fermi states and boundary states.

Let us now turn to a four-band model. We consider a Dirac-like psH-symmetric model described by

$$\begin{aligned} \mathcal{H}_{\text{psH}}^{4b} = &\left\{ t \left[ \cos(k_x) + \cos(k_y) - 2 \right] + t_z \left[ \cos(k_z) - \cos(k_0) \right] \right\} \left( \frac{2}{\sqrt{3}} \Upsilon^{14} + \sqrt{\frac{2}{3}} \Upsilon^{15} \right) \\ &+ i\lambda_{Ixy} \sin(k_y)(\Upsilon^3 + \Upsilon^4) + i\lambda_{Ixx} \sin(k_x)(\Upsilon^9 + \Upsilon^{10}) \\ &+ im_I' \sin(k_z) \left[ \cos(k_x) - \cos(k_y) \right] (\Upsilon^7 - \Upsilon^{12}), \end{aligned} \tag{19}$$

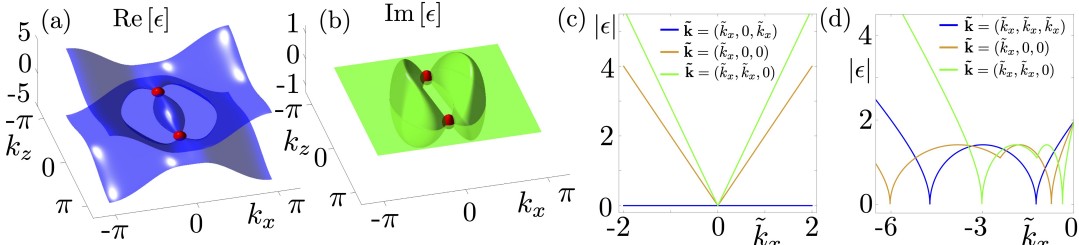

Figure 4: Real (a) and imaginary (b) part of band structure for $\mathcal{H}_{\text{psH}}^{4b}$ in Eq. (19) along $k_x = k_y$ and $k_z$. Note that along this particular cut, both bands are doubly degenerate. Red points indicate non-defective EPs at $\mathbf{k} = (0,0,\pm k_0)$. Also here, it is clear that the absolute value of the eigenvalues depend linearly on the momentum components close to the non-defective EP at $\mathbf{k} = (0,0,k_0)$ [panel (c)], while a fractional behavior is observed around defective EPs, exemplified around $\mathbf{k} = \left[\pi/2, \pi/2, \arccos\left(2-\sqrt{2}\right)\right]$ in panel (d). Here we set $t = t_z = \lambda_{Ixx} = \lambda_{Ixy} = 1$, $m_I' = -0.27$ and $k_0 = \pi/2$.

where $t$, $t_z$, $\lambda_{Ixy}$, $\lambda_{Ixx}$, and $m_I'$ are real-valued parameters. This model is a pseudo-Hermitian generalization of the tight-binding model studied in Ref. [55]. The trivial band touching points for obtaining the null form of the traceless part of $\mathcal{H}_{\text{psH}}^{4b}$ in Eq. (19) are located at $\mathbf{k} = \mathbf{k}_{\text{psH}} = (0,0,\pm k_0)$. Right at these points and on lines connecting these points, constraints for realizing EP4s, summarized in Table 3 and given in Eq. (7), are also satisfied. Hence, points at $\mathbf{k} = \mathbf{k}_{\text{psH}}$ in our psH-symmetric model are non-defective EP4s. We present these nodal points (red spheres) in the band structure of $\mathcal{H}_{\text{psH}}^{4b}$ in Fig. 4. The real part of the spectra (a) shows that two arc-shaped surfaces with (non)zero real (imaginary) parts are terminated by the non-defective EPs as well as the defective exceptional lines. These surfaces are the aforementioned NH bulk real-Fermi surfaces. While our model hosts bulk Fermi surfaces, boundary states are unstable, similar to their Hermitian counterparts [55].

Similar to the case of the two-band model, we investigate the behavior of the dispersion and eigenvector coalescence near the non-defective EP. Linearizing $\mathcal{H}_{\text{psH}}^{4b}$ around $\mathbf{k} = \mathbf{k}_{\text{psH}}$ yields the following eigensystem,

$$\epsilon_{\text{psH}}^{\pm} = \pm \sqrt{t_z^2 \tilde{k}_z^2 - \lambda_{xx}^2 \tilde{k}_x^2 - \lambda_{xy}^2 \tilde{k}_y^2}, \tag{20}$$

$$\psi_{\text{psH}}^{1,\pm} = \begin{pmatrix} \frac{i\left(-t_z \tilde{k}_z \mp \sqrt{t_z^2 \tilde{k}_z^2 - \lambda_{xx}^2 \tilde{k}_x^2 - \lambda_{xy}^2 \tilde{k}_y^2}\right)}{\lambda_{xx}\tilde{k}_x + i\lambda_{xy}\tilde{k}_y} \\ 0 \\ 0 \\ 1 \end{pmatrix}, \tag{21}$$

$$\psi_{\text{psH}}^{2,\pm} = \begin{pmatrix} 0 \\ \frac{i\left(-t_z \tilde{k}_z \mp \sqrt{t_z^2 \tilde{k}_z^2 - \lambda_{xx}^2 \tilde{k}_x^2 - \lambda_{xy}^2 \tilde{k}_y^2}\right)}{\lambda_{xx}\tilde{k}_x + i\lambda_{xy}\tilde{k}_y} \\ 1 \\ 0 \end{pmatrix}, \tag{22}$$

where $\tilde{\mathbf{k}} = \mathbf{k} - \mathbf{k}_{\text{psH}}$ and we note that in this linearized regime, the eigenvalues are doubly degenerate. The dispersion is again linear in the momentum components, as displayed for the respective absolute values in Fig. 4(c). To contrast this, Fig. 4(d) shows the dispersion around a defective EP, which is of the square-root type. Approaching the origin along the defective

EPs defined by $t_z^2 \tilde{k}_z^2 - \lambda_{xx}^2 \tilde{k}_x^2 - \lambda_{xy}^2 \tilde{k}_y^2$, the eigenvectors become

$$\lim_{t_z \tilde{k}_z \to \sqrt{\lambda_{xx}^2 \tilde{k}_x^2}} \lim_{\tilde{k}_y \to 0} \psi_{\mathrm{psH}}^{1,\pm} = \begin{pmatrix} -\frac{\mathrm{i}\sqrt{\lambda_{xx}^2 \tilde{k}_x^2}}{\lambda_{xx} \tilde{k}_x} \\ 0 \\ 0 \\ 1 \end{pmatrix}, \tag{23}$$

$$\lim_{t_z \tilde{k}_z \to \sqrt{\lambda_{xx}^2 \tilde{k}_x^2}} \lim_{\tilde{k}_y \to 0} \psi_{\mathrm{psH}}^{2,\pm} = \begin{pmatrix} 0 \\ -\frac{\mathrm{i}\sqrt{\lambda_{xx}^2 \tilde{k}_x^2}}{\lambda_{xx} \tilde{k}_x} \\ 1 \\ 0 \end{pmatrix}, \tag{24}$$

making it clear that we only have two linearly independent eigenvectors when approaching the non-defective EP and that the point looks as if it is defective also when $\tilde{k}_x \to 0$. Yet again, when approaching the origin from another direction, e.g.,

$$\lim_{\tilde{k}_y \to 0} \lim_{\tilde{k}_z \to 0} \psi_{\mathrm{psH}}^{1,\pm} = \begin{pmatrix} \pm\mathrm{sign}(\tilde{k}_x) \\ 0 \\ 0 \\ 1 \end{pmatrix}, \tag{25}$$

$$\lim_{\tilde{k}_y \to 0} \lim_{\tilde{k}_z \to 0} \psi_{\mathrm{psH}}^{2,\pm} = \begin{pmatrix} 0 \\ \pm\mathrm{sign}(\tilde{k}_x) \\ 1 \\ 0 \end{pmatrix}, \tag{26}$$

none of the eigenvectors coalesce when $\tilde{k}_x \to 0$. Just as in the previous case, this is reflected in the corresponding Jordan decomposition, which along the defective EPs reads

$$S_{\mathrm{psH}}^{4b} = \begin{pmatrix} -\frac{\mathrm{i}\sqrt{\lambda_{xx}^2 \tilde{k}_x^2 + \lambda_{xy}^2 \tilde{k}_y^2}}{\lambda_{xx} \tilde{k}_x + \mathrm{i}\lambda_{xy} \tilde{k}_y} & -\frac{\mathrm{i}}{\lambda_{xx} \tilde{k}_x + \mathrm{i}\lambda_{xy} \tilde{k}_y} & 0 & 0 \\ 0 & 0 & -\frac{\sqrt{\lambda_{xx}^2 \tilde{k}_x^2 + \lambda_{xy}^2 \tilde{k}_y^2}}{\lambda_{xx} \tilde{k}_x + \mathrm{i}\lambda_{xy} \tilde{k}_y} & -\frac{\mathrm{i}}{\lambda_{xx} \tilde{k}_x + \mathrm{i}\lambda_{xy} \tilde{k}_y} \\ 0 & 0 & 1 & 0 \\ 1 & 0 & 0 & 0 \end{pmatrix}, \tag{27}$$

$$J_{\mathrm{psH}}^{4b} = \begin{pmatrix} 0 & 1 & 0 & 0 \\ 0 & 0 & 0 & 0 \\ 0 & 0 & 0 & 1 \\ 0 & 0 & 0 & 0 \end{pmatrix}. \tag{28}$$

The matrix $S_{\mathrm{psH}}^{4b}$ has a singularity at the origin, corresponding exactly to the non-defective EP. Thus, the Jordan decomposition exhibits an instability exactly at the non-defective EP.

## 3 Searching for the coexistence of ONPs and defective EPs

So far, we have explored how the presence of one of the $\mathcal{PT}$, $\mathcal{CP}$ or psH symmetries leads to the general coexistence of defective and non-defective EPs. Now, we instead address whether ONPs may also exist in NH spectra. For this purpose, we lift (discrete) symmetry restrictions such that some (or all) $d_\mu$'s have real and imaginary parts. We emphasize that in this situation, in contrast to the EPs in the previous sections, non-defective band degeneracies are generally

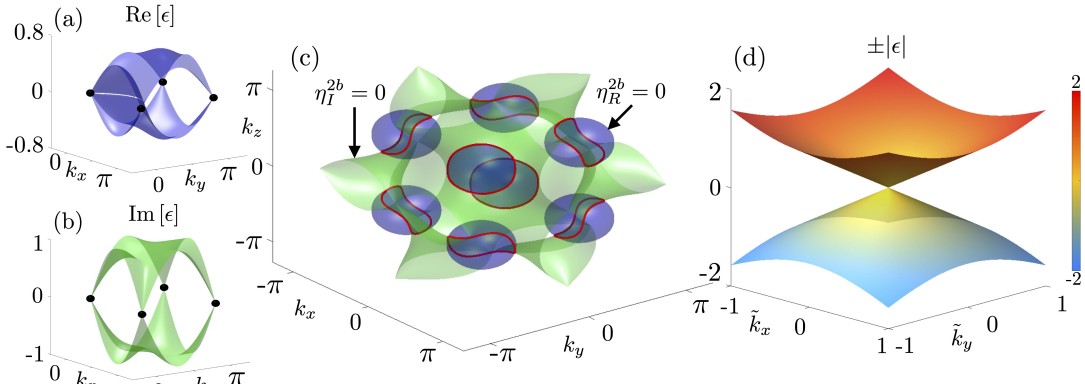

Figure 5: Real (a) and imaginary (b) part of band structure for $\mathcal{H}_{\mathrm{ONP}}^{2b}$ in Eq. (29) along $k_z = \pi$. (c) Solutions of the real (blue spheres) and imaginary (green manifold) parts of $\eta^{2b}$ in Eq. (5). Red closed lines in (c) present the intersection of these manifolds and correspond to defective EPs. Black points in panels (a) and (b) indicate ONPs. For visibility purposes, merely a corner of the Brillouin zone is shown in panels (a) and (b). Panel (d) displays that the absolute values of the eigenvalues depend linearly on the momentum components close to the ONP located at the origin.

unstable to small perturbations.[4] This is because solving $\mathbf{d} = 0$ generally requires satisfying $n^2 - 1$ complex constraints. Consequently, the appearance of ONPs in NH models is vulnerable to the fine-tuning of parameters, and therefore, these systems may not exhibit experimental signatures different from those in generic NH systems. Nevertheless, we show in the following that this setting provides a platform to observe ONPs.

To illustrate our idea, we introduce a two-band model given by

$$
\begin{aligned}
\mathcal{H}_{\mathrm{ONP}}^{2b} = {} & \sin(k_x)\big[1/2 + i\cos(k_y)\big]\Upsilon^1 \\
& + \sin(k_y)\big[1/2 + i\cos(k_z)\big]\Upsilon^2 \\
& + \sin(k_z)\big[1/2 + i\cos(k_x)\big]\Upsilon^3 .
\end{aligned}
\tag{29}
$$

Fig. 5 exhibits the real (a) and imaginary (b) part of the energy dispersion of this system along $k_z = \pi$. The black points in these panels mark real and imaginary band touching points located at $k_{x,y,z}^{\mathrm{ONP}} = \pm n\pi$. Even though at these momenta the traceless part of $\mathcal{H}_{\mathrm{ONP}}^{2b}$ becomes a null matrix, we emphasize that $\mathbf{k} = \mathbf{k}^{\mathrm{ONP}}$ indicates the location of ONPs and not non-defective EPs. The reason for this statement lies in the fact that the criterion for the emergence of non-defective EPs is not satisfied, i.e., no defective EPs reside close to $\mathbf{k}^{\mathrm{ONP}}$. This can be seen from Fig. 5(c) in which we present defective EPs, red curves, as the intersection between $\mathrm{Re}[\eta^{2b}] = 0$, blue spheres, and $\mathrm{Im}[\eta^{2b}] = 0$, green manifold. Fig. 5(c) reveals that defective EPs do not cross $\mathbf{k} = \mathbf{k}^{\mathrm{ONP}}$ and thus the black points at $\mathbf{k}^{\mathrm{ONP}}$ indeed correspond to ONPs. Consequently, $H_{\mathrm{ONP}}^{2b}$ is diagonalizable at $\mathbf{k} = \mathbf{k}^{\mathrm{ONP}}$ and in any small neighborhood surrounding the point; see Fig. 5(d) for a momentum expansion of eigenvalues, with $\tilde{\mathbf{k}} = \mathbf{k}$, around the origin.

Aside from these ONPs in the momentum space, introducing models that host boundary states connecting ONPs in NH systems is theoretically feasible. We present an example of such a model in Appendix D.

---

[4]More precisely, twofold degeneracies are protected by composite symmetries consisting of multiple symmetry operations [56]. Respecting all these symmetry operations might be easily violated upon introducing perturbations.

# 4 Footprints of non-Hermitian degeneracies in experiments and their plausible physical implications

$\mathcal{PT}$-symmetric models have been extensively realized in various experimental setups, including superconducting quantum processors [57], optics [58, 59], photonics [60–62], acoustics [63], electronic circuits [64, 65] and flying atoms [59]. While all of these experiments report observing footprints of $\mathcal{PT}$ (un)broken phases in their measurements, experimental constraints, e.g., limited ranges of parameters, restrict the measurements to be performed close to one EP and usually away from the non-defective EPs.[5] For this reason, most of the above-mentioned experiments do not confirm the distinct behavior of different types of EPs in their observations.

The quantum setup reported in Ref. [59] is one of the exceptions, where the accessible parameter space allows for the appearance of the EP pairs. The two-channel model is obtained from the density matrix formalism upon imposing approximations on different parameters. Aside from the model, the authors of Ref. [59] also reported the measured transmission spectra and compared those with their theoretical model. Looking at the transmission spectra in Hermitian and NH systems (see Figs. 2 and 3 in Ref. [59]), we recognize distinct features. They report a two-peak (single-peak) structure of the transmission spectra in the $\mathcal{PT}$ (un)broken phases. The peaks in the unbroken phase are zero-centered, resembling the zero slope of the intersection connecting non-defective EPs to defective EPs. In contrast, in the (effective) Hermitian model, transmission spectra for each channel exhibit single peaks keeping opposite distances from zero. This reflects the linear dispersion around regular degeneracies. In the $\mathcal{PT}$-broken phase, a two-peak structure with non-zero centers of the maxima for the transmission spectra is reported. These observations demonstrate how different types of non-Hermitian degeneracies can give rise to distinct features in different observables.

The microwave experiments with metallic mesh 3D photonic crystals have also realized $\mathcal{PT}$-symmetric models [66] with chains of nodal lines. Exceptional chains have recently been observed in mechanical systems [67], where the intersection of these EP lines represents non-defective EPs. Sometimes, these non-defective degeneracies are protected by additional mirror symmetries; see Refs. [68, 69] for details.

In addition to the setups mentioned above, numerous studies on heterostructures report the occurrence of EPs in these systems [6, 70–72]. However, to the best of our knowledge, no record of non-defective degeneracies is reported. Nevertheless, further engineering of the structure of these systems to maintain the symmetries discussed here may allow for the emergence of non-defective degeneracies and the experimental realization of our findings in these setups.

Aside from realizing NH degeneracies, the occurrence of these degeneracies in the spectrum may give rise to exotic responses. The NH anomalous currents observed in odd spatial dimensions exemplify these interesting responses. It has been shown that NH (non)interacting systems with ONPs, when coupled to gauge fields, e.g., electromagnetic fields, exhibit anomalous currents different from their Hermitian analog [5, 73, 74]. For instance, the NH chiral magnetic effect, in contrast to its Hermitian counterpart, may find room to emerge in equilibrium in $\mathcal{PT}$-symmetric systems [73].

---

[5]Note that in these systems, EPs emerge in pairs; see Sec. 2 for more details.

# 5 Conclusion

Despite the intense focus on NH systems in recent studies, the possibility of realizing different types of EPs has hitherto been overlooked. The present work shows that two different types of EPs, dubbed defective and non-defective EPs, may coexist in various setups of physical importance. We show that non-defective EPs are stabilized by certain symmetries, including $\mathcal{PT}$, $\mathcal{CP}$, psH, and time-reversal symmetry. To resolve the confusion in the current literature, where non-defective EPs are mixed up with ONPs, we have in this work introduced a clear criterion to distinguish these concepts. We also highlight this difference in example models.

Our systems, especially $\mathcal{PT}$-symmetric models, are experimentally feasible as they can be implemented in experimental optical setups with balanced gain and loss [75]. As exploring the role of EPs in $\mathcal{PT}$-symmetric optical systems has already unraveled many interesting phenomena [60, 76–79], we expect that finding possibilities to detect ONPs and (non-)defective EPs may also pave the way to advance applications of Hermitian and NH topological properties in various fields of research.

# Acknowledgments

**Funding information** M.S. is supported by the Swedish Research Council (VR) and the Wallenberg Academy Fellows program of the Knut and Alice Wallenberg Foundation. L.R. acknowledges the support from the Knut and Alice Wallenberg foundation under grant no. 2017.0157.

# A Bases matrices for two-, three-, and four-band systems

### Basis matrices for two-band systems

The basis matrices for two-band systems are Pauli matrices, which read

$$\Upsilon^1 = \begin{pmatrix} 0 & 1 \\ 1 & 0 \end{pmatrix}, \quad \Upsilon^2 = \begin{pmatrix} 0 & -i \\ i & 0 \end{pmatrix}, \quad \Upsilon^3 = \begin{pmatrix} 1 & 0 \\ 0 & -1 \end{pmatrix}. \tag{A.1}$$

### Basis matrices for three-band systems

The basis matrices for three-band systems are the Gell-Mann matrices that span the Lie algebra of the SU(3) group,

$$\Upsilon^1 = \begin{pmatrix} 0 & -i & 0 \\ i & 0 & 0 \\ 0 & 0 & 0 \end{pmatrix}, \qquad \Upsilon^2 = \begin{pmatrix} 0 & 0 & -i \\ 0 & 0 & 0 \\ i & 0 & 0 \end{pmatrix}, \tag{A.2}$$

$$\Upsilon^3 = \begin{pmatrix} 0 & 0 & 0 \\ 0 & 0 & -i \\ 0 & i & 0 \end{pmatrix}, \qquad \Upsilon^4 = \begin{pmatrix} 0 & 1 & 0 \\ 1 & 0 & 0 \\ 0 & 0 & 0 \end{pmatrix}, \tag{A.3}$$

$$\Upsilon^5 = \begin{pmatrix} 0 & 0 & 1 \\ 0 & 0 & 0 \\ 1 & 0 & 0 \end{pmatrix}, \qquad \Upsilon^6 = \begin{pmatrix} 0 & 0 & 0 \\ 0 & 0 & 1 \\ 0 & 1 & 0 \end{pmatrix}, \tag{A.4}$$

$$\Upsilon^7 = \begin{pmatrix} 1 & 0 & 0 \\ 0 & -1 & 0 \\ 0 & 0 & 0 \end{pmatrix}, \qquad \Upsilon^8 = \begin{pmatrix} \frac{1}{\sqrt{3}} & 0 & 0 \\ 0 & \frac{1}{\sqrt{3}} & 0 \\ 0 & 0 & -\frac{2}{\sqrt{3}} \end{pmatrix}. \tag{A.5}$$

**Basis matrices for four-band systems**

The basis matrices for four-band systems are the generalized Gell-Mann matrices that span the Lie algebra of the SU(4) group,

$$\Upsilon^1 = \begin{pmatrix} 0 & -i & 0 & 0 \\ i & 0 & 0 & 0 \\ 0 & 0 & 0 & 0 \\ 0 & 0 & 0 & 0 \end{pmatrix}, \qquad \Upsilon^2 = \begin{pmatrix} 0 & 0 & -i & 0 \\ 0 & 0 & 0 & 0 \\ i & 0 & 0 & 0 \\ 0 & 0 & 0 & 0 \end{pmatrix}, \tag{A.6}$$

$$\Upsilon^3 = \begin{pmatrix} 0 & 0 & 0 & -i \\ 0 & 0 & 0 & 0 \\ 0 & 0 & 0 & 0 \\ i & 0 & 0 & 0 \end{pmatrix}, \qquad \Upsilon^4 = \begin{pmatrix} 0 & 0 & 0 & 0 \\ 0 & 0 & -i & 0 \\ 0 & i & 0 & 0 \\ 0 & 0 & 0 & 0 \end{pmatrix}, \tag{A.7}$$

$$\Upsilon^5 = \begin{pmatrix} 0 & 0 & 0 & 0 \\ 0 & 0 & 0 & -i \\ 0 & 0 & 0 & 0 \\ 0 & i & 0 & 0 \end{pmatrix}, \qquad \Upsilon^6 = \begin{pmatrix} 0 & 0 & 0 & 0 \\ 0 & 0 & 0 & 0 \\ 0 & 0 & 0 & -i \\ 0 & 0 & i & 0 \end{pmatrix}, \tag{A.8}$$

$$\Upsilon^7 = \begin{pmatrix} 0 & 1 & 0 & 0 \\ 1 & 0 & 0 & 0 \\ 0 & 0 & 0 & 0 \\ 0 & 0 & 0 & 0 \end{pmatrix}, \qquad \Upsilon^8 = \begin{pmatrix} 0 & 0 & 1 & 0 \\ 0 & 0 & 0 & 0 \\ 1 & 0 & 0 & 0 \\ 0 & 0 & 0 & 0 \end{pmatrix}, \tag{A.9}$$

$$\Upsilon^9 = \begin{pmatrix} 0 & 0 & 0 & 1 \\ 0 & 0 & 0 & 0 \\ 0 & 0 & 0 & 0 \\ 1 & 0 & 0 & 0 \end{pmatrix}, \qquad \Upsilon^{10} = \begin{pmatrix} 0 & 0 & 0 & 0 \\ 0 & 0 & 1 & 0 \\ 0 & 1 & 0 & 0 \\ 0 & 0 & 0 & 0 \end{pmatrix}, \tag{A.10}$$

$$\Upsilon^{11} = \begin{pmatrix} 0 & 0 & 0 & 0 \\ 0 & 0 & 0 & 1 \\ 0 & 0 & 0 & 0 \\ 0 & 1 & 0 & 0 \end{pmatrix}, \qquad \Upsilon^{12} = \begin{pmatrix} 0 & 0 & 0 & 0 \\ 0 & 0 & 0 & 0 \\ 0 & 0 & 0 & 1 \\ 0 & 0 & 1 & 0 \end{pmatrix}, \tag{A.11}$$

$$\Upsilon^{13} = \begin{pmatrix} 1 & 0 & 0 & 0 \\ 0 & -1 & 0 & 0 \\ 0 & 0 & 0 & 0 \\ 0 & 0 & 0 & 0 \end{pmatrix}, \qquad \Upsilon^{14} = \begin{pmatrix} \frac{1}{\sqrt{3}} & 0 & 0 & 0 \\ 0 & \frac{1}{\sqrt{3}} & 0 & 0 \\ 0 & 0 & -\frac{2}{\sqrt{3}} & 0 \\ 0 & 0 & 0 & 0 \end{pmatrix}, \tag{A.12}$$

$$\Upsilon^{15} = \begin{pmatrix} \frac{1}{\sqrt{6}} & 0 & 0 & 0 \\ 0 & \frac{1}{\sqrt{6}} & 0 & 0 \\ 0 & 0 & \frac{1}{\sqrt{6}} & 0 \\ 0 & 0 & 0 & -\sqrt{\frac{3}{2}} \end{pmatrix}. \tag{A.13}$$

# B  Derivation of constraints to find EP*ns*

Here we briefly summarize how to derive Eqs. (5, 6, 7) in the main text, where these constraints were originally derived in Ref. [42]. There, it was shown that the characteristic polynomial of an *n*-band NH matrix can be expressed in terms of its determinant and traces. Indeed, for

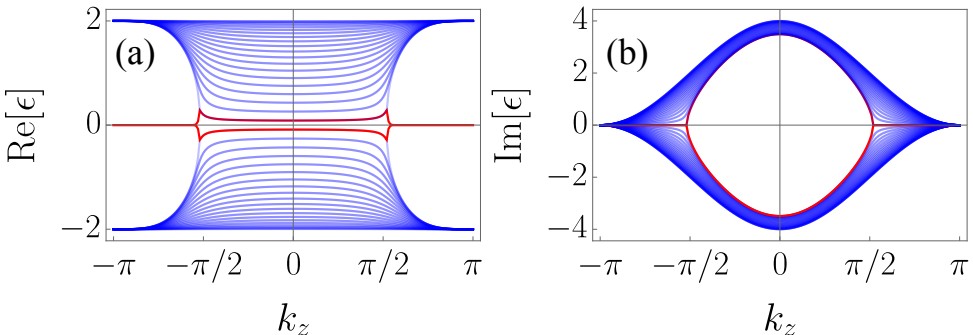

Figure 6: Real (a) and imaginary (b) part of spectra for Eq. (12) with open boundary condition in the $y$ direction and $k_x = 0$. Red lines present boundary states. Here we set $t = V = \lambda_0 = 1.0$.

two-, three- and four-band matrices, these polynomials read

$$F^{2b}(\lambda) = \lambda^2 - \text{tr}[\mathcal{H}]\lambda + \det[\mathcal{H}] = 0, \tag{B.1}$$

$$F^{3b}(\lambda) = \lambda^3 - \text{tr}[\mathcal{H}]\lambda^2 + \frac{\text{tr}[\mathcal{H}]^2 - \text{tr}[\mathcal{H}^2]}{2}\lambda - \det[\mathcal{H}] = 0, \tag{B.2}$$

$$F^{4b}(\lambda) = \lambda^4 - a\lambda^3 + b\lambda^2 - c\lambda + d = 0, \tag{B.3}$$

where

$$a = \text{tr}[\mathcal{H}], \quad b = \frac{\text{tr}[\mathcal{H}]^2 - \text{tr}[\mathcal{H}^2]}{2}, \quad d = \det[\mathcal{H}], \tag{B.4}$$

$$c = \frac{\text{tr}[\mathcal{H}]^3 - 3\text{tr}[\mathcal{H}]\text{tr}[\mathcal{H}^2] + 2\text{tr}[\mathcal{H}^3]}{6}, \tag{B.5}$$

and $\lambda$ are the eigenvalues of $\mathcal{H}$.

To find degeneracies, the discriminant $D[\mathcal{H}]$ of these characteristic polynomials should be set to zero. The discriminants read

$$D[\mathcal{H}^{2b}] = \text{tr}[\mathcal{H}]^2 - 4\det[\mathcal{H}], \tag{B.6}$$

$$D[\mathcal{H}^{3b}] = -\frac{1}{27}[4(\eta^{3b})^3 + (\nu^{3b})^2], \tag{B.7}$$

$$D[\mathcal{H}^{4b}] = \frac{1}{27}[4(\eta^{4b})^3 - (\nu^{4b})^2], \tag{B.8}$$

where $\eta^{3b}$ and $\nu^{3b}$, and $\eta^{4b}$ and $\nu^{4b}$ are given in Eqs. (6) and (7) in the main text, respectively. From here, we immediately see that setting the discriminants in Eqs. (B.6) and (B.7) to zero gives us the constraints in Eqs. (6) and (7) in the main text, respectively. In the case of the four-band model, we note that for all roots of the discriminant to coincide, not only $\eta^{4b} = 0$ and $\nu^{4b} = 0$ need to be satisfied but also $\kappa^{4b} = 0$, where $\kappa^{4b}$ is defined in Eq. (7) in the main text. We refer to Ref. [42] for a more detailed discussion on this point.

## C  Spectra of the two-band model with open boundary condition

In addition to the properties of momentum-dependent spectra for $\mathcal{H}^{2b}_{\mathcal{PT}}$ in Fig. 2, we present the real (a) and imaginary (b) part of the energy dispersion with open boundary condition in the $y$ direction plotted in Fig. 6. The figure exhibits boundary states, red lines, well-separated from the bulk states (blue lines) when $|k_z| > \pi/2$.

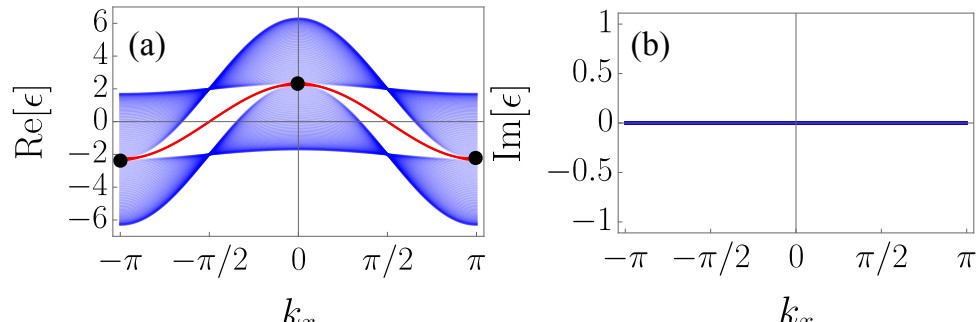

Figure 7: Real (a) and imaginary (b) part of spectra for Eq. (D.1) with open boundary condition in the $y$ direction and at $k_z = 0$. Red lines present boundary states. Black points mark ONPs. Here we set $t = V = 1.0$, $\lambda_0 = 2.3$.

## D   Realizing Hermitian boundary states in non-Hermitian systems

Here, we present a non-Hermitian tight-binding model hosting Hermitian boundary states, which connects ONPs with zero imaginary parts.

Our two-band model Hamiltonian reads

$$
\begin{aligned}
\mathcal{H}^{2b}_{edg} = {} & \lambda_0 \cos(k_x)\Upsilon^0 - iV\left[1 - \cos(k_z)\right]\Upsilon^1 \\
& + \left[2V\cos(k_y) - 2t\cos(k_x)\right]\Upsilon^1 \\
& - 2t\sin(k_y)\Upsilon^2 - 2t\sin(k_z)\Upsilon^3,
\end{aligned} \tag{D.1}
$$

where $\lambda_0$, $t$ and $V$ are real-valued coupling constants. Along $k_z = 0$, the above Hamiltonian is fully Hermitian. As a result, nodal points, which live on the $(k_x, k_y)$ plane, are band-touching points with zero imaginary parts. For instance, these ordinary nodal points appear at $\mathbf{k}_{\text{ONPs}} = (\pm n\pi, \pm n\pi, 0)$ with $n \in \mathbb{Z}$ when $t = V$. Considering the open boundary condition along the $y$ axis and at $k_z = 0$ results in mid-gap boundary states, red lines in Fig. 7(a), which connects $\mathbf{k}_{\text{ONPs}}$.

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
