# Peer review of "Symmetry-protected exceptional and nodal points in non-Hermitian systems"

_SciPost Physics, doi:SciPost Phys. 15, 200 (2023)_

## Round 1 · Referee Report · Anonymous (Referee 1) · 2023-6-19

Strengths

1.

Report

Dear Editor, hereby I submit my report on "Symmetry-protected exceptional and nodal points in non-Hermitian systems" by Sharareh Sayyad, Marcus Stalhammar, Lukas Rodland, Flore K. Kunst.

The authors studies non-Hermitian degeneracies in non-Hermitian systems subjected to symmetries. In particular, the authors address defective, non-defective and ordinary Hermitian degeneracies. Of great importance is the recipe to distinguish the three types of degeneracies in non-Hermitian setups, which is of importance towards the detecting of exceptional points and their potential applications. Furthermore, I would like to highlight that the manuscript is well-written and the messages clearly exposed. For these reasons I recommend its acceptance in SciPost after addressing the minor comments listed below, which might be useful to further improve the manuscript:

  1. In page 10 below equation 29, in the sentence starting with "This can be seen from Fig. 4(b)..". I have the feeling that it should be Fig.4(c).

  2. I would recommend the authors to add additional references, highlighting the immense efforts of several groups. For instance, see following suggestions:

2.1 Together with Refs. [1-5], the authors could incorporate the first non-Hermitian studies on a consented matter matter junction: JETP Lett. 94, 693 (2012); Scientific Reports 6, 21427 (2016).

2.2 Together with Refs. [6-19]: Phys. Rev. B 99, 165145 (2019); Phys. Rev. Research 4, L012006 (2022)

2.3 Together with Refs.[28-38], I recommend adding: Phys. Rev. B 97, 014512 (2018); Phys. Rev. B 107, 104515 (2023); Phys. Rev. B 107, 115146 (2023); Proc. Int. Conf. on Strongly Correlated Electron Systems (SCES2019) (Physical Society of Japan, 2019) Chap. 30, p. 011098.

  1. In section 4, the authors discuss about identifying exceptional points in experiments. Even though the authors address some relevant systems, I believe their manuscript could have a bigger impact if they also discuss non-Hermitian systems emerging from material junctions. The material junctions are very relevant in transport experiments, see e.g., Nat. Rev. Mater. 6, 944 (2021), and represent a natural platform for non-Hermitian physics. Examples of non-Hermitian physics in junctions have been initially reported in JETP Lett. 94, 693 (2012); Scientific Reports 6, 21427 (2016). Later, more interesting studies were reported which further support the importance of non-Hermitian physics in material junctions, see e.g., Phys. Rev. B 87, 235421 (2013).; Phys. Rev. Research 1, 012003(R) (2019); Phys. Rev. B 103, 235438 (2021); J. Phys.: Condens. Matter 35, 254002 (2023).

Please, let me know if I can be of further assistance.

  • validity: -
  • significance: -
  • originality: -
  • clarity: -
  • formatting: -
  • grammar: -

Author:  Sharareh Sayyad  on 2023-09-13  [id 3977]

(in reply to Report 1 on 2023-06-19)
Category:
answer to question
correction

We would like to thank the referee for the submitted report. We believe that the feedback has helped us improve the manuscript, in particular, when in relation to applications. Attached, we provide answers to the questions raised by the referees with the original question included in teal.

Attachment:

reply_referee1.pdf

---

## Round 1 · Referee Report · Anonymous (Referee 2) · 2023-6-23

Strengths

1- The technical computations are clearly exposed and easy to follow. 2- The studies of degeneracies in non-Hermitian systems has clearly become very relevant for experiments. This paper brings a clearer mathematical understanding to the nuances of their properties. 3- A formal characterization of the NDEP would be appreciated.

Weaknesses

1- In my opinion, the paper fails to properly introduce the different types of degeneracies in non-Hermitian systems. A short,mathematical definition (with an example for each) in Introduction or Appendix would greatly simplify the reading of the paper. I , in particular, would stress the importance of a more explicit definition of defective vs non-defective exceptional points early in the paper (it is barely half a sentence currently). 2- Similarly, the authors fail to emphasize the relevance of the distinction between defective and non-defective EP, in terms of physics.

Report

The paper "Symmetry-protected exceptional and nodal points in non-Hermitian systems" gives generic criterion of existence for defective and non-defective high-dimensional exceptional points in general n bands models. Given the relevance of the non-Hermitian descriptions of experiments and the possibilities offered by exceptional points. the manuscript appears to be relevant to a general public.

While the paper is globally well presented, it is hard to follow, especially for non specialists. I also think there are several important typos. Consequently, I would only recommend the publication in SciPost after modifications.

Given the mistakes I found, I also strongly recommend a careful rereading of the manuscript as it is not impossible I missed others.

I had a few questions, in addition to the changes I would like to see listed below. 1- Non-defective exceptional point are characterized as diagonalizable degenerate points in the neighborhood of which defective EP exists. Is there a form of (topological) invariant/signature one can derive to characterize them instead? 2- These NDEP split two manifolds/lines of DEP. Given this structure, a property should abruptly change when following these lines of DEP through the NDEP. At first sight, I expect that some properties of the defective eigenstates dramatically change (typically, handedness in the example you have) . Is this intuition correct and general? Can you take advantage of that to explain the resilience of the NDEP in high dimensions?

Requested changes

1- Give concrete and explicit mathematical definitions of the different type of degeneracies discussed in the manuscript, preferably in Section II (or in an Appendix for examples) and stressing the differences. 2-End of page 4, you discuss $TRS^\dagger$. You claim that "it enforces all symmetric parts of d to be 0". It does not seem to be the case for $d_{x, I}$, $d_{y, R}$ and $d_{z, I}$ which should be symmetric. Following that it seems that the corresponding result in the table is incorrect. 3- The limits in Eq. 15 appear ill defined: the limit on $k_y$ is only well defined if $k_z^2 > k_x^2$ and then $k_z$ is sent to 0 while $k_x$ is kept finite. The limit on $k_z$ does not appear to be necessary to show the desired property: fixing $k_y^2 = k_z^2 - k_x^2$ is enough. 4- Fig. 2a: the spectrum is not symmetric under kx -> -kx. Given the form and the parameter, there seems to be a mistake. 5- Could you clarify the reason of the double< degeneracy in your 4 band model. 6- The spectrum in Fig. 4 6- Could you develop/clarify and illustrate in Section 4 the experimental signature of the presence of non-defective EP

Minor points 7- Fig. 2 and 3 should be made larger. In both cases, c) and d) are barely understandable. Given that both Figs describe traceless models with only 2 effective bands (with the double degeneracy for Fig. 4), I would recommend plotting a pcolor map (or something similar) of one of the bands only for all of these graphs. If the authors want to stress square root profiles (or other), then showing a cut would probably be enough. 8- Eq. 17 and 18 are valid for $\tilde{k}_y = \sqrt{k_z^2-k_x^2}$. It is not very clearly specified. Also, I seem to find the denominator to be $\tilde{k}_x + \tilde{k}_z$, though it might be a question of convention. 9- Eq. 21 and 22: I also have a minus sign in front of kz. 10- Eq.23 and 24: The two limits are again not really necessary.

  • validity: ok
  • significance: high
  • originality: good
  • clarity: ok
  • formatting: excellent
  • grammar: excellent

Author:  Sharareh Sayyad  on 2023-09-13  [id 3978]

(in reply to Report 2 on 2023-06-23)

We would like to thank the referee for the submitted report. We believe that the feedback has helped us improve the manuscript, in particular, when in relation to applications. Attached, we provide answers to the questions raised by the referee with the original question included in teal.

Attachment:

reply_referee2.pdf

---

## Round 2 · Referee Report · Anonymous (Referee 3) · 2023-9-13

Report

Second report on "Symmetry-protected exceptional and nodal points in non-Hermitian systems" by Sharareh Sayyad, Marcus Stålhammar, Lukas Rodland, Flore K. Kunst.

The authors have addressed all my previous comments and their answers to the concerns of the other referees seems convincing. Therefore, I recommend the acceptance of this manuscript for publication in SciPost Physics.

Requested changes

None

---

## Round 2 · Referee Report · Anonymous (Referee 4) · 2023-10-13

Report

This is my second report on "Symmetry-protected exceptional and nodal points in non-Hermitian systems" by Sharareh Sayyad, Marcus Stålhammar, Lukas Rodland, Flore K. Kunst.

The authors have addressed my comments, and corrected the typos in their article. Given my previous report, I recommend the acceptance of this manuscript for publication in SciPost Physics.

---

## Editorial Decision

published